# Robust variability of grid cell properties within individual grid modules enhances encoding of local space

William T Redman[1,2]*[†], Santiago Acosta-Mendoza[1†], Xue-Xin Wei[3,4,5,6], Michael J Goard[7,8,9]*

[1]Interdepartmental Graduate Program in Dynamical Neuroscience, University of California, Santa Barbara, Santa Barbara, United States; [2]Intelligent Systems Center, Johns Hopkins University Applied Physics Lab, Laurel, United States; [3]Department of Neuroscience, The University of Texas at Austin, Austin, United States; [4]Center for Learning and Memory, The University of Texas at Austin, Austin, United States; [5]Center for Perceptual Systems, The University of Texas at Austin, Austin, United States; [6]Center for Theoretical and Computational Neuroscience, The University of Texas at Austin, Austin, United States; [7]Department of Psychological and Brain Sciences, University of California, Santa Barbara, Santa Barbara, United States; [8]Department of Molecular, Cellular, and Developmental Biology, University of California, Santa Barbara, Santa Barbara, United States; [9]Neuroscience Research Institute, University of California, Santa Barbara, Santa Barbara, United States

*For correspondence:
will.redman@jhuapl.edu (WTR);
michael.goard@lifesci.ucsb.edu
(MJG)

[†]These authors contributed
equally to this work

**Competing interest:** The authors
declare that no competing
interests exist.

**Reviewing Editor:** Adrien
Peyrache, McGill University,
Canada

## eLife Assessment

This **valuable** study examines the variability in spacing and direction of entorhinal grid cells, providing **convincing** evidence that such variability helps disambiguate locations within an environment. This study will be of interest to neuroscientists working on spatial navigation and, more broadly, on neural coding.

**Abstract** Although grid cells are one of the most well-studied functional classes of neurons in the mammalian brain, whether there is a single orientation and spacing value per grid module has not been carefully tested. We analyze a recent large-scale recording of medial entorhinal cortex to characterize the presence and degree of heterogeneity of grid properties within individual modules. We find evidence for small, but robust, variability and hypothesize that this property of the grid code could enhance the encoding of local spatial information. Performing analysis on synthetic populations of grid cells, where we have complete control over the amount heterogeneity in grid properties, we demonstrate that grid property variability of a similar magnitude to the analyzed data leads to significantly decreased decoding error. This holds even when restricted to activity from a single module. Our results highlight how the heterogeneity of the neural response properties may benefit coding and opens new directions for theoretical and experimental analysis of grid cells.

## Introduction

The discovery of grid cells in medial entorhinal cortex (MEC; *Hafting et al., 2005*) has led to considerable experimental and computational work aimed at identifying their origin (*Guanella and Verschure, 2006*; *Fuhs and Touretzky, 2006*; *Blair et al., 2007*; *Couey et al., 2013*; *Dordek et al., 2016*; *Cueva*

and Wei, 2018; Banino et al., 2018; Weber and Sprekeler, 2018; Sorscher et al., 2019; Khona et al., 2022; Sorscher et al., 2023) and their function (McNaughton et al., 2006; Solstad et al., 2006; Rolls et al., 2006; Burak and Fiete, 2009; de Almeida et al., 2009; Kubie and Fox, 2015; Bush et al., 2015; Ormond and McNaughton, 2015; Mallory et al., 2018). The organization of grid cells into discrete modules (Stensola et al., 2012; Gu et al., 2018), with grid properties (grid spacing and orientation) clustered within module, but not between modules, has fundamentally shaped this research. For instance, the increasing size and spacing of grid modules along the dorsal-ventral axis of MEC, by discontinuous jumps of a near constant ratio, has been argued to be optimal for encoding local spatial information (Wei et al., 2015; Stemmler et al., 2015) when grid cell activity across all modules is integrated together (Fiete et al., 2008; Sreenivasan and Fiete, 2011; Mathis et al., 2012a; Wei et al., 2015; Stemmler et al., 2015). This is despite the fact that the individual neurons in hippocampus, a downstream target of the MEC, receive inputs from only a portion of the dorsal-ventral axis (van Strien et al., 2009). The modularity of the grid system has also been proposed to simplify the wiring necessary for generating continuous attractor dynamics (Fuhs and Touretzky, 2006; Guanella and Verschure, 2006; Burak and Fiete, 2009; Couey et al., 2013), a computational mechanism theorized to underlie grid cells function that enjoys considerable experimental support (Yoon et al., 2013; Dunn et al., 2015; Gu et al., 2018; Gardner et al., 2019; Trettel et al., 2019; Gardner et al., 2022).

Much of this prior theoretical work has made the additional assumption that grid cell properties are identical, up to a phase shift, within a single module (Sreenivasan and Fiete, 2011; Mathis et al., 2012a; Wei et al., 2015; Stemmler et al., 2015; Dorrell et al., 2023). However, as experimentalists have been aware, the distributions of measured orientation and spacing show non-zero variability (Stensola et al., 2012; Yoon et al., 2013; Gardner et al., 2022). This could be due to finite recording time, neurophysiological noise, and/or the sensitivity of numerical methods used to fit grid properties. Alternatively, this variability could reflect underlying inhomogeneity, at a fine scale, within modules.

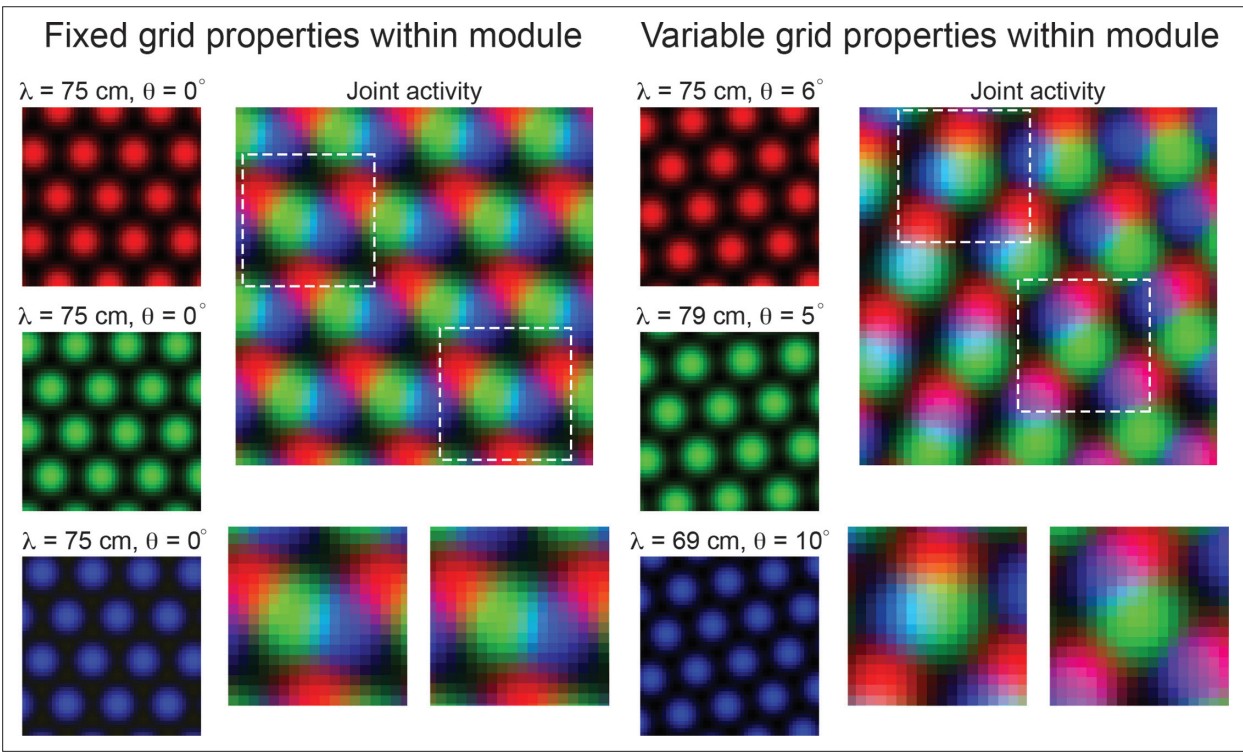

**Figure 1.** Variability in grid cell properties within a module leads to enhanced encoding of local space. When the activity of three idealized grid cells, all with the same grid spacing and orientation, are considered, the periodicity of the responses limits the amount of information conveyed about local space (Left column – 'Fixed grid properties within module'). That is, there are multiple locations in physical space with identical population level activity. However, when three grid cells with variable grid spacing and orientation (in the realm of what is measured within individual grid modules – see Results), their joint activity contains considerably more information (Right column – 'Variable grid properties within module'). This benefit of spatial inhomogeneity is expected to increase with larger populations of grid cells. Dashed squares in the joint activity map are enlarged below.

Despite the fundamental way in which grid cell function has been informed by their modular organization, to the best of our knowledge, no characterization of the degree and robustness of variability in grid properties within individual modules has been performed.

If robust variability of grid properties does exist, then it is possible that this heterogeneity could be exploited to encode additional information about local space, reducing the requirement of integration across multiple grid modules (*Figure 1*). In particular, when all grid cells in a module have the same grid orientation and spacing, the joint population activity has a translational invariance that persists, even as larger populations of grid cells are considered (*Figure 1*, 'Fixed grid properties within module'). Thus, it is not possible to resolve spatial position from population activity. In contrast, if grid cells within a module have variability in their grid orientation and spacing, then – over a finite area – the translational invariance of the population activity is broken. In such a case, distinct patterns of population activity emerge in distinct locations of space (*Figure 1*, 'Variable grid properties within module'). As an example, the blue and green grid cells in *Figure 1* show the most overlap in the upper left half of the arena (denoted by cyan pixels), while the red and blue grid cells show the most overlap in the lower right portion of the arena (denoted by purple pixels). This is despite the fact that the variability in spacing and orientation is only on the order of a few centimeters and degrees, respectively. We expect that these patterns should become more complex and contain more information that could be used to disentangle spatial location as more grid cells are considered.

In this paper, we perform detailed analysis of recent state-of-the-art MEC electrophysiological recordings, which were made publicly available (*Gardner et al., 2022*). We characterize the variability of grid orientation and spacing within individual modules and find evidence for small, but robust, variability. This variability is present whether a single value of grid orientation and spacing is assigned to each grid cell, or whether grid distortion (*Derdikman et al., 2009*; *Krupic et al., 2015*) is taken into account by considering three grid orientations and spacings independently. Performing similar analysis on recent normative recurrent neural network (RNN) models of grid cells (*Sorscher et al., 2019*), we find the presence of comparable variability. This provides another possible explanation for why these RNN models have been found to capture MEC grid cell response profiles (*Nayebi et al., 2021*). To assess the functional implications of this heterogeneity, we perform simulation experiments with synthetic, noisy grid cell populations, where we have complete control over the distribution of grid orientation and spacing. We find that the variability in grid cell orientation and spacing, at a similar degree as present in the data we analyze, leads to lower decoding error of local space when using the activity of a *single* module.

Taken together, our results challenge a frequently made assumption in the theoretical literature and support a growing understanding of the spatial information encoded by grid cell populations (*Diehl et al., 2017*; *Ismakov et al., 2017*; *Dunn et al., 2017*; *Ginosar et al., 2023*). Additionally, our results encourage consideration of the broader benefits that multiple modules may provide, beyond the encoding of local space (*Hawkins et al., 2018*; *Klukas et al., 2020*; *Rueckemann et al., 2021*).

## Results
### Robust differences in grid cell properties within individual modules
To determine the extent to which variability of grid properties in individual modules exists, and to what extent this variability is a robust property of the grid code, we analyzed previously published MEC recordings (*Gardner et al., 2022*), which include tens to hundreds of grid cells simultaneously recorded. This allows us to characterize the distribution of grid properties within a single grid module, to an extent not possible with other data sets.

For each grid cell, we compute the grid spacing ($\lambda$) and orientation ($\theta$) by measuring properties associated with the six hexagonally distributed peaks of the spatial autocorrelogram (SAC), as traditionally performed (*Sargolini et al., 2006*; *Figure 2A*; see Materials and methods). For clarity, we begin by focusing on a single module, recorded from an open field environment (recording identifier: Rat R, Day 1, Module 2 – R12). This module was picked for its long recording time (approximately 130 min of recorded activity, as compared to the other open field recordings that have 30 min or less of recorded activity) and for its large number of simultaneously recorded putative grid cells ($N = 168$). In this module, we find that grid cells with high grid scores ($> 0.85$; $N = 74$) have a range of grid orientation and spacing (example cells shown in *Figure 2B*; distributions across module shown in *Figure 2E*

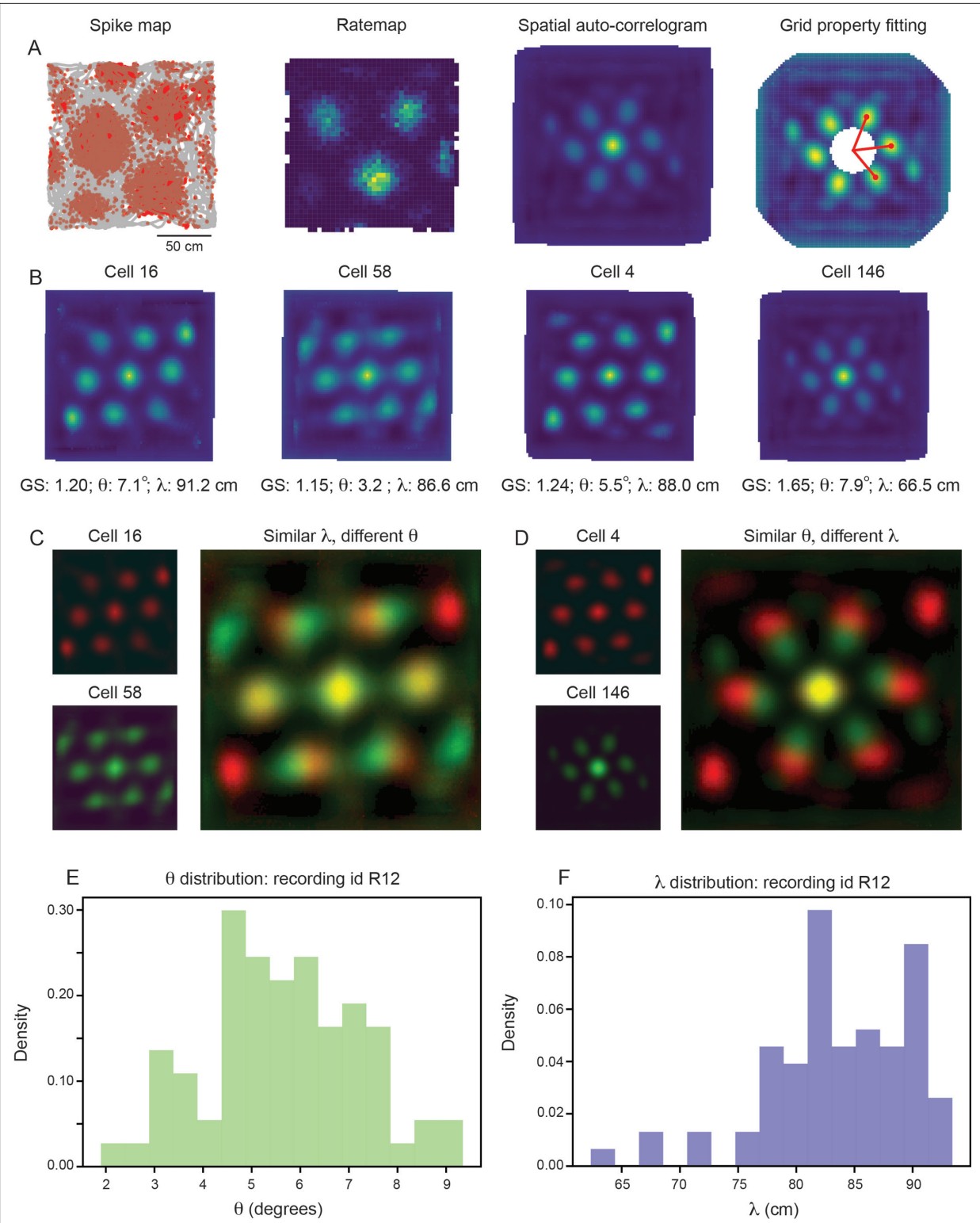

**Figure 2.** Grid properties are variable within a single grid module (recording ID R12). (**A**) Overview of the standard procedure used to calculate the grid spacing and orientation of a given grid cell. First, spike maps are computed by identifying the location of the animal at the time of each spike. Gray line denotes the trajectory of the rat, red dots denote locations of spikes. A rate map is constructed by binning space and normalizing by the amount of time the rat spent in each spatial bin. A spatial autocorrelogram (SAC) is computed and, after the center peak is masked out (white pixels in the center of the spatial autocorrelogram – leading to change in color scale), the grid properties are fit by measuring the length and angle of the three peaks closest to 0°. (**B**) Example grid cells from the same module (recording ID R12), with estimated grid score, orientation (θ), and spacing (λ). (**C, D**) SAC overlaid for

*Figure 2 continued on next page*

*Figure 2 continued*

two pairs of grid cells (from **B**); one pair with different $\theta$ and similar $\lambda$ (**C**) and the other with similar $\theta$ and different $\lambda$ (**D**). (**E, F**) Distribution of $\theta$ (**E**) and $\lambda$ (**F**) across all grid cells with grid score >0.85 (N=74).

The online version of this article includes the following figure supplement(s) for figure 2:

**Figure supplement 1.** Variability in grid spacing within a single module exists when computing $\lambda$ directly from the rate maps.

*and F*), with $\lambda$ ranging from approximately 65 cm to 90 cm and $\theta$ ranging from approximately 2° to 9°. Overlaying the SACs of pairs of grid cells with similar grid spacing and different grid orientation (*Figure 2C*) or vice versa (*Figure 2D*) enables visualization of the extent of this variability.

Because individual grid fields can be cut-off by the boundaries of the environment, it is possible that computing the grid spacing from the SAC (which considers all grid fields) could lead to an underestimate of $\lambda$ for some grid cells. To verify that the broad distribution of grid spacing that we see within the same module is not due to the specifics of the SAC, we recomputed the grid spacing for all grid cells directly from their rate maps, considering only the three grid fields closest to the center of the environment (*Figure 2—figure supplement 1A*; see Materials and methods). We find that while the grid spacing estimated from the SAC tends to be larger than the grid spacing estimated from the rate maps, a similarly broad range of $\lambda$ is again present (*Figure 2—figure supplement 1B, C*).

To assess whether the heterogeneity of grid properties present in a single grid module is a robust feature or attributable to noise (either in the recording or the grid property fitting procedure), we measure the variability in grid orientation and spacing within a single grid cell and between pairs of grid cells. If the heterogeneity is explainable by noise then we expect that the within-cell variability will be of the same magnitude as the between-cell variability. In contrast, if the heterogeneity is a robust feature of the grid code, then we expect the within-cell variability will be significantly smaller than the between-cell variability.

To measure the within- and between-cell variability, we split the recording into evenly spaced 30 s bins, randomly assigning each temporal bin to one of two equal length halves and computing the grid properties for each half of the recording (*Figure 3A*; see Materials and methods). We set inclusion criteria to filter out cells that do not have consistent hexagonally structured SACs across splits of the recording (see Materials and methods). Although these requirements are strict (see *Figure 3—figure supplement 1* for percent of cells rejected), they set a conservative estimate on the amount of grid property variability, ensuring that we do not artificially inflate the variability due to inclusion of unreliable grid cells. We found that the length of the temporal bin used to split the data does not have a large impact on the percentage of cells accepted by this criteria (*Figure 3—figure supplement 2*; see Materials and methods).

We find that the distribution, across 100 random shuffles of the data into two halves, of withincell variability of grid orientation and spacing is more concentrated around 0 than the between-cell variability (*Figure 3B and C*). Comparing the average within- and between-cell variability of grid spacing and orientation reveals that nearly all of the grid cells that passed the criteria for inclusion ($N = 82$) exhibit more between-cell than within-cell variability (*Figure 3D and E*): 95.1% grid cells for orientation ($\langle\overline{\Delta\theta}_{\text{within}}\rangle = 1.2°$, $\langle\overline{\Delta\theta}_{\text{between}}\rangle = 2.0°$; $\langle\cdot\rangle$ denotes mean across cells) and 100% of grid cells for spacing ($\langle\overline{\Delta\lambda}_{\text{within}}\rangle = 1.9\,\text{cm}$, $\langle\overline{\Delta\lambda}_{\text{between}}\rangle = 6.9\,\text{cm}$). A Wilcoxon-Signed-Rank Test indicates that between-cell variability is significantly greater than within-cell variability, for both orientation and spacing ($\Delta\theta : p < 0.001$, $\Delta\lambda : p < 0.001$). We perform a number of control analyses to confirm that our results are robust, finding that: (1) the average grid field width does not significantly affect the between-cell variability (*Figure 3—figure supplement 3*), (2) the amount of between-cell variability does not change significantly across the recording duration (*Figure 3—figure supplement 4*), (3) the between-cell variability is not significantly impacted by the boundaries of the arena (*Figure 3—figure supplement 5*), and (4) the amount of between-cell variability is not driven by the presence of cells that have both head direction and grid cell like tuning 'conjunctive' cells (*Sargolini et al., 2006*; *Figure 3—figure supplement 6*).

In keeping with convention, the reported grid cell properties are the average of those computed for each of the three independent axes in the SAC (Axis 1: aligned to ≈ 0°; Axis 2: aligned to ≈ 60°; Axis 3: aligned to ≈ 60° *Stensola et al., 2015*). To ensure that this averaging is not contributing to greater between-cell variability, we repeated the analysis above, restricting ourselves to each axis separately. The results again demonstrate that grid properties are significantly more robust within-cell

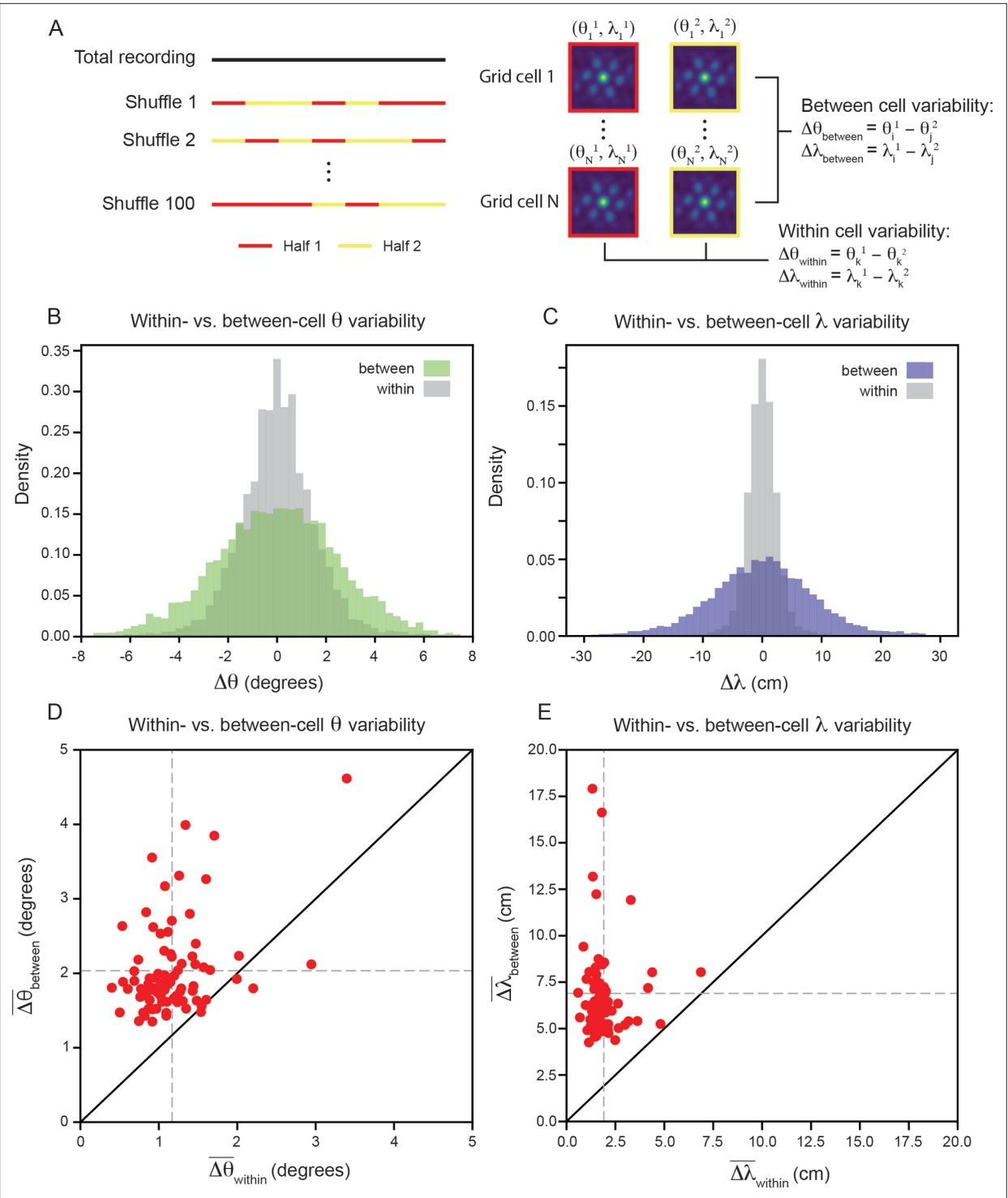

**Figure 3.** Variability of grid properties is a robust feature of individual grid module (recording ID R12). (**A**) Schematic overview of approach used to compute the between- and within-cell variability of grid orientation and spacing. (**B, C**) Distribution of within- and between-cell variability of $\theta$ and $\lambda$, respectively. Note that the distribution is across all 100 random shuffles of the data into two halves. (**D**) Average within-cell variability of grid orientation ($\overline{\Delta\theta}_{\text{within}}$), compared to average between-cell variability of grid spacing ($\overline{\Delta\theta}_{\text{between}}$). (**E**) Same as (**D**), but for $\lambda$. 1 cell was excluded from (**E**) for visualization ($\overline{\Delta\lambda}_{\text{between}} > 20\,\text{cm}$), but was included in non-parametric statistical analysis. For (**D, E**), $N = 82$.

The online version of this article includes the following figure supplement(s) for figure 3:

**Figure supplement 1.** Accepted and rejected cells across all grid modules.

*Figure 3 continued on next page*

*Figure 3 continued*

**Figure supplement 2.** Bin length does not affect the percent of cells accepted for analysis.

**Figure supplement 3.** Average field width does not play a significant role in explaining the between-cell variability in spacing.

**Figure supplement 4.** Between-cell variability does not significantly change across recording time.

**Figure supplement 5.** Variability in grid spacing and orientation is not significantly affected by location of arena boundaries.

**Figure supplement 6.** Conjunctive head direction grid cells do not significantly affect the variability in grid orientation and spacing.

**Figure supplement 7.** Path integrating recurrent neural networks (RNNs) that develop grid cells exhibit variability in spacing and orientation that scales with recurrent weight sparsity.

than between-cell (*Figure 4A and B*). For each axis, the average between-cell variability for every cell was significantly higher than the average within-cell variability (*Figure 4C and D*), as reported by the Wilcoxon-Signed-Rank Test for orientation (p < 0.001, for all three axes) and for spacing (p < 0.001, for all three axes).

Having demonstrated that grid cell properties are robustly heterogeneous in a single module, we proceed to analyze the remaining recordings in the data set ($N = 420$ cells across 8 modules; one module in the data set had no cells that passed our criteria; *Gardner et al., 2022*). Although there are differences across recordings, we find larger between- than within-cell variability for grid orientation and spacing is present across all recordings (*Figure 5A and B*; 80.0% of grid cells have greater between- than within-cell variability for orientation, $\langle \overline{\Delta\theta}_{\text{within}} \rangle = 1.4°$ and $\langle \overline{\Delta\theta}_{\text{between}} \rangle = 1.9°$; 87.9% of grid cells have greater between- than within-variability for spacing, $\langle \overline{\Delta\lambda}_{\text{within}} \rangle = 1.7 \, \text{cm}$ and $\langle \overline{\Delta\lambda}_{\text{between}} \rangle = 3.6 \, \text{cm}$). A Wilcoxon-Signed-Rank Test finds that these differences are significant ($\Delta\theta : p < 0.001; \Delta\lambda : p < 0.001$). We do not find evidence suggesting that the within-cell variability is influenced by grid score (*Figure 5C and D*; linear regression for $\theta : R^2 = 0.03, p = 0.55$ Wald Test; linear regression for $\lambda : R^2 = 0.07, p = 0.14$ Wald Test).

To understand whether the observed grid cell variability may be a heretofore unknown property of MEC computational models, we trained recurrent neural networks (RNN) to perform path integration. These RNN models have previously been shown to develop grid responses (*Banino et al., 2018*; *Cueva and Wei, 2018*; *Sorscher et al., 2019*) and have been argued to develop continuous attractor network structure (*Sorscher et al., 2023*), making them normative models for MEC function. We find that the RNNs develop grid responses with a distribution of grid spacing and orientation (*Figure 3—figure supplement 7A-D*), much like the neural data we analyzed (*Figure 2*). Changing the sparsity of the recurrent layer by increasing the strength of weight decay regularization reduces the variability (*Figure 3—figure supplement 7E, F*).

Taken together, our analysis of large-scale MEC recordings demonstrates that grid cells in the same grid module do not have a single grid spacing and orientation, but instead have a restricted range of values around the module mean. This property is a robust feature that cannot be explained by noise from the recording or fitting procedures and emerges in existing normative models of MEC.

## Variability in grid properties within individual modules improves the encoding of local space

The variability of grid cell properties within individual grid modules, while statistically significant, is small in magnitude. Can a computational benefit in the encoding of local space be gained from this level of inhomogeneity? How sensitive might such a computational benefit be to the exact amount of variability present?

To address these questions, we generate populations of synthetic grid cells (see Materials and methods), where we have complete control over the number of grid cells and their firing properties. For simplicity, we assume that all grid cells in our population have grid orientation and spacing sampled from Gaussian distributions, with means $\hat{\lambda}$ and $\hat{\theta}$ and standard deviations $\sigma_\lambda$ and $\sigma_\theta$, respectively. Assigning each grid cell in our population a grid orientation and spacing, we are able to generate 'ideal' rate maps (*Solstad et al., 2006*). Sampling from a Poisson process on these ideal rate maps, we generate noisy grid cell rate maps (*Figure 6A*). Using a simple linear decoder (see Materials and methods), we can examine how decoding error of local space is affected by the number of grid cells in the population and the amount of variability ($\sigma_\lambda$ and $\sigma_\theta$).

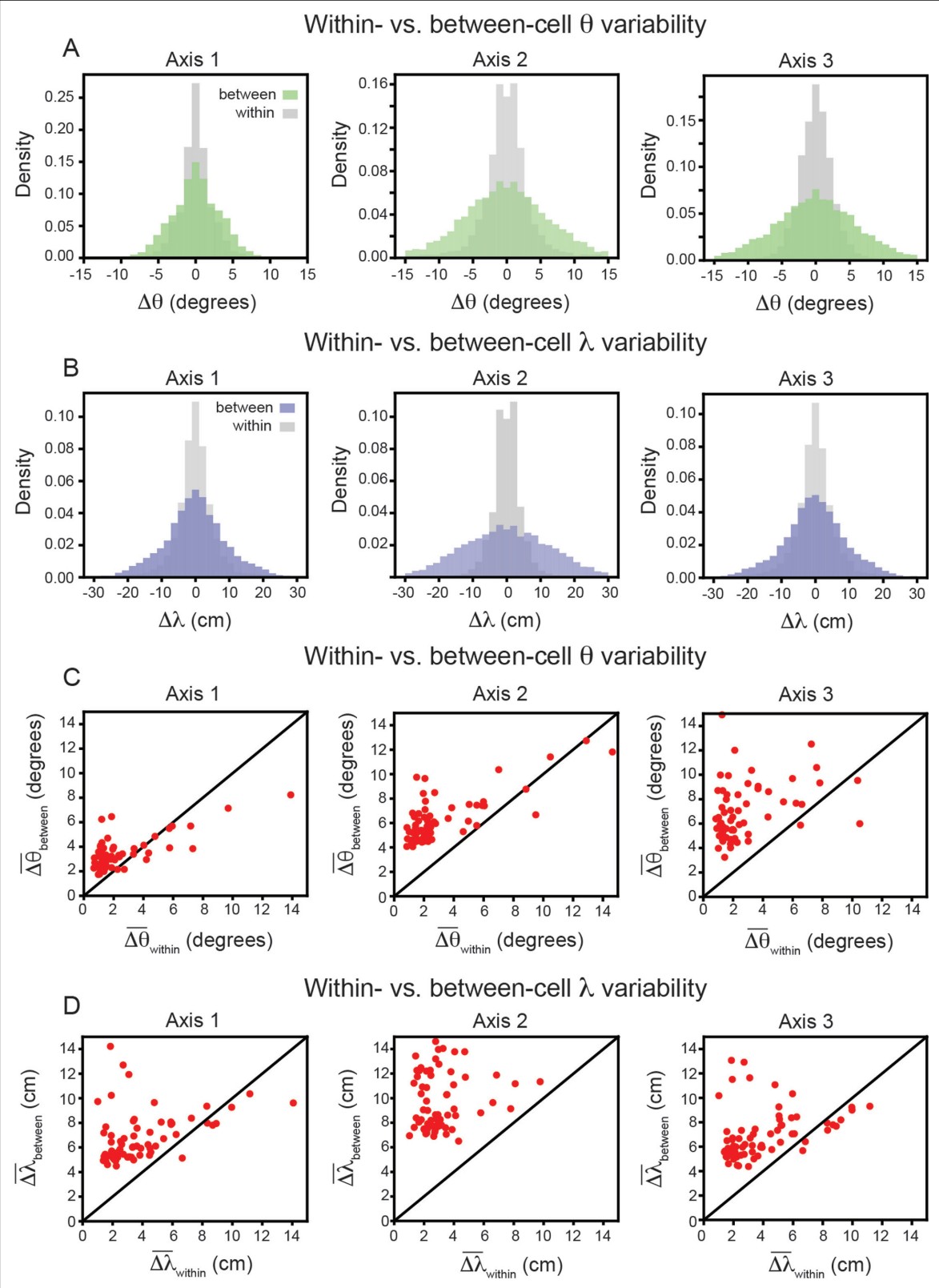

**Figure 4.** Variability of grid properties, restricted to the same axis, is a robust feature of individual grid module (recording ID R12). Same analysis as in *Figure 3B–E*, but for variability measured on each axis independently. (**A**) Distribution of orientation variability ($\Delta\theta$) for each grid axis. (**B**) Distribution of spacing variability ($\Delta\lambda$) for each grid axis. (**C**) Within- vs. between-cell orientation variability ($\overline{\Delta\theta}$) for each grid axis. (**D**) Within- vs. between-cell spacing variability ($\overline{\Delta\lambda}$) for each grid axis. For visualization, we exclude a small number of cells that were outside the axes limits, including 2, 5, and 10 cells for Axes 1–3, respectively (**C**); and 3, 4, and 3 cells for Axes 1–2, respectively (**D**); these cells were included in non-parametric statistical analyses.

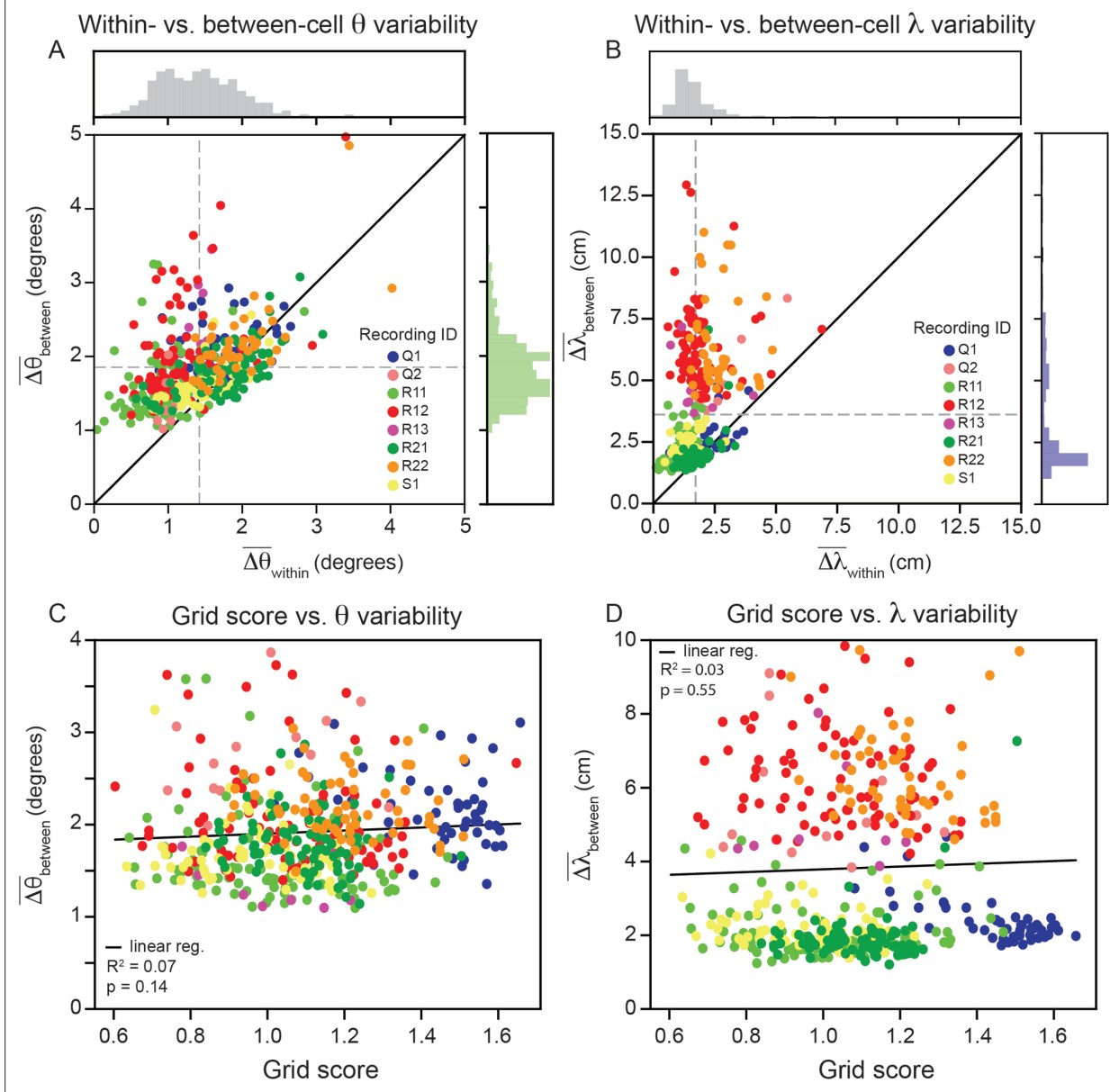

**Figure 5.** Within module grid property variability is a robust feature across modules. (**A**) Average within-cell variability of grid orientation ($\overline{\Delta\theta}_{\text{within}}$), compared to average between-cell variability of grid orientation ($\overline{\Delta\theta}_{\text{between}}$) for each cell ($N = 420$) across 8 modules (cells colored by their corresponding recording ID). The histogram above the plot shows the distribution of $\overline{\Delta\theta}_{\text{within}}$ and the histogram to the right shows the distribution of $\overline{\Delta\theta}_{\text{between}}$. (**B**) Same as (**A**), but for grid spacing. For visualization, 5 cells are excluded ($\overline{\Delta\lambda}_{\text{between}} > 15\,\text{cm}$), but are included in non-parametric statistical analyses. Dashed gray lines show the population mean. (**C, D**) Average within cell variability of $\theta$ and $\theta$ (respectively), as a function of grid score. For visualization, 3 and 22 cells are excluded from (**C, D**), respectively, but are included in statistical analyses. Black solid line is linear regression, with $R^2$ and $p$-value reported above.

We begin by investigating the decoding capabilities of a synthetic module with properties similar to that of the experimentally recorded module that was analyzed in detail (recording identifier R12; *Figures 2 and 3*). We therefore set $\hat{\lambda} = 85\,\text{cm}$ and $\hat{\theta} = 6°$. To determine an appropriate value for $\sigma_\lambda$ and $\sigma_\theta$, we subtract the mean between-cell variability by the mean within-cell variability ($\langle\overline{\Delta\theta}_{\text{between}}\rangle - \langle\overline{\Delta\theta}_{\text{within}}\rangle = 2.0° - 1.2°$ and $\langle\overline{\Delta\lambda}_{\text{between}}\rangle - \langle\overline{\Delta\lambda}_{\text{within}}\rangle = 6.9\,\text{cm} - 1.9\,\text{cm} = 5.0\,\text{cm}$). We interpret these values as the amount of variability in grid spacing and orientation that is not due to noise. Therefore, we set $\sigma_\theta = 1°$ and $\sigma_\lambda = 5\,\text{cm}$. This amount of variability leads to similar sampled distributions of $\lambda$ and $\theta$ as was found in the real data (compare *Figure 6B and C* with *Figure 2E and F*).

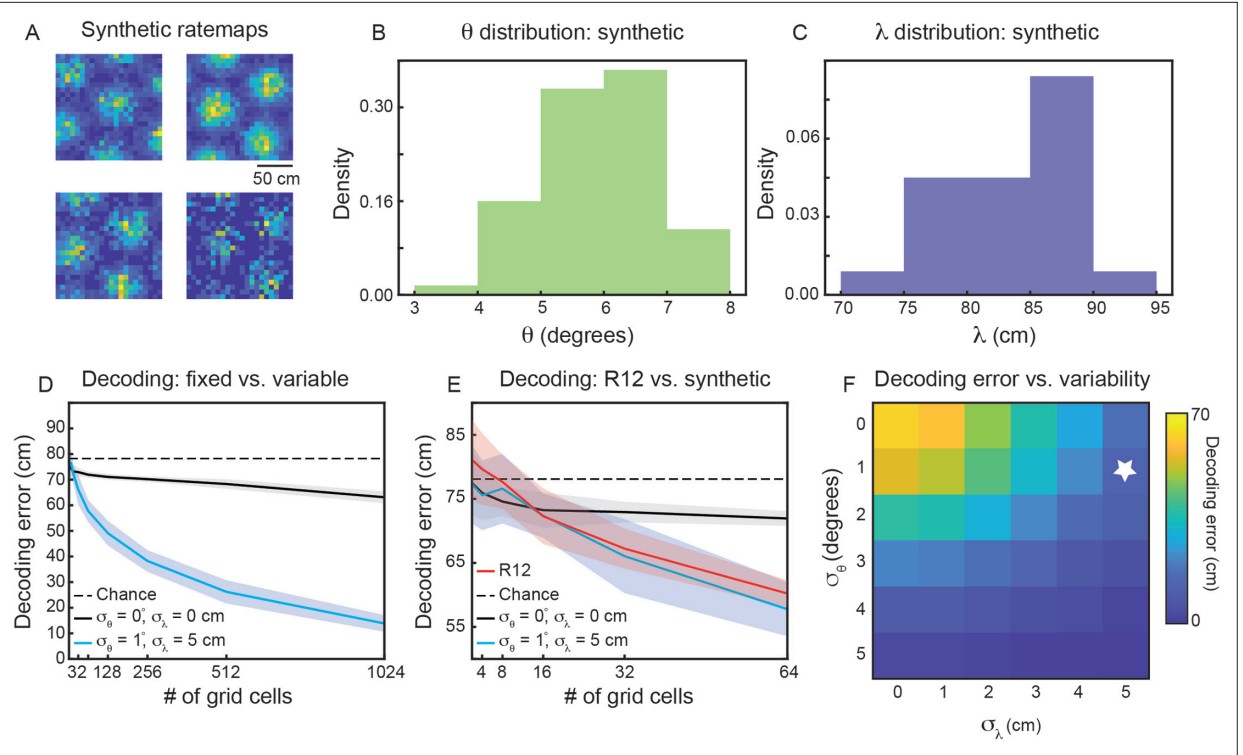

**Figure 6.** Variability in grid properties enables improved decoding of local space from the activity of grid cells within a single module. (**A**) Example noisy grid cell rate maps generated from a Poisson process. The size of the square arena is set to 1.5 m ×1.5 m to be consistent with what was used in the experimental set-up analyzed (*Gardner et al., 2022*). (**B, C**) Distribution of sampled grid spacing and orientation from synthetic population, when using $\hat{\lambda} = 85\,\text{cm}$, $\hat{\theta} = 6°$, $\sigma_\lambda = 5\,\text{cm}$ cm, and $\sigma_\theta = 1°$; compare to the distribution measured from real data (*Figure 2E and F*). (**D**) Decoding error, as a function of grid cell population size, with populations having either no variability in grid properties (black line) or variability similar to what was present in the data analyzed (blue line). The solid line is the mean across 25 independent grid cell populations and the shaded area is ± standard deviation of the 25 independent populations. The dashed black line shows chance level decoding error. (**E**) Decoding error for synthetic populations and real data for up to $N = 64$ cells (red line). (**F**) Decoding error, over a grid of $\sigma_\theta$ and $\sigma_\lambda$ values, for populations of $N = 1024$ grid cells. White star denotes values used in (**D**).

The online version of this article includes the following figure supplement(s) for figure 6:

**Figure supplement 1.** Variability in grid properties improves decoding of local space for grid modules with different mean grid spacings.

**Figure supplement 2.** Variability in grid properties improves decoding of local space for multiple modules, when the modules have integer multiple mean spacing.

Applying the linear decoder to populations of synthetic grid cells, we find that decoding error decreases as the number of grid cells is increased (*Figure 6D*, blue line). When $N = 1024$, the decoding error is ≈ 20 cm, substantially better than random decoding (*Figure 6D*, dashed black line). As expected, this result is not seen in a synthetic population of grid cells with zero variability in grid properties ($\sigma_\theta = 0°$ and $\sigma_\lambda = 0\,\text{cm}$; *Figure 6D*, black line). In particular, while the restricted number of grid firing fields enables a decoding error smaller than chance, the population with fixed grid properties exhibits little change in decoding error with increasing numbers of grid cells. This demonstrates that the improved encoding is specific to populations with inhomogeneity in their grid spacing and orientation.

To validate that the synthetic population is a reasonable surrogate to the experimentally recorded data, we perform the same decoding analysis on grid cells from recording ID R12 (see Materials and methods). As the real data is limited in the number of cells, we are only able to compare up to populations of $N = 64$ (*Figure 6E*). However, in this restricted range, we find agreement between the synthetic population with grid property variability and the recorded data (*Figure 6E*, compare red and blue lines). This supports our hypothesis that the observed amounts of inhomogeneity in grid spacing and orientation can lead to improved decoding of local space.

To determine the extent to which the decrease in decoding error depends the exact variability of grid spacing and orientation, we perform a grid search over 36 pairs of $(\sigma_\lambda, \sigma_\theta)$ values in [0°, 1°, ...

,5°]×[0 cm, 1 cm, ... , 5 cm]. As expected, when the variability of both grid properties is large, we find nearly 0 cm decoding error of local space (*Figure 6E*, bottom right). However, there is additionally a range of $\sigma_\lambda$ and $\sigma_\theta$ values that lead to improved decoding, including values smaller than the values matched to the experimental data (*Figure 6E*, above and to the left of the white star). We perform the same grid search on synthetic populations of grid cells with different spacings, $\hat{\lambda} = 50\,cm$ and $\hat{\lambda} = 100\,cm$ (*Figure 6—figure supplement 1A, B*), finding again that the decoding error drops below that of the fixed population with sufficient grid property variability. For the synthetic module with smaller spacing ($\hat{\lambda} = 50\,cm$), the existence of more grid fields in the 1.5 m × 1.5 m arena leads to greater complexity of interference patterns (*Figure 1*), enabling the same amount of variability in grid spacing and orientation to lead to a sharper decrease in decoding error. Additionally, for the synthetic module with smaller spacing, a given value of $\sigma_\lambda$ is a larger percentage of $\hat{\lambda}$, as compared to the synthetic module with larger spacing.

Finally, we consider how decoding within a single module compares to decoding across two modules. When the two modules are consecutive (e.g. modules 1 and 2), their mean grid spacing, $\hat{\lambda}_1$ and $\hat{\lambda}_2$, has been experimentally found to be related via $\hat{\lambda}_2 \approx \sqrt{2}\hat{\lambda}_1$ (*Stensola et al., 2012*). In such a case, decoding activity from populations with fixed grid properties leads to nearly 0 cm error (*Figure 6—figure supplement 2A*, black line). The addition of variability in grid properties does not significantly change the behavior of the decoding (*Figure 6—figure supplement 2A*, blue line). This suggests that small amounts of inhomogeneity may not disrupt the previously achieved theoretical bounds on decoding from multiple grid modules (*Mathis et al., 2012a*; *Mathis et al., 2012b*; *Stemmler et al., 2015*; *Wei et al., 2015*). However, if the two modules being decoded from are non-consecutive (e.g. modules 1 and 3) then the mean grid spacing can be related by an integer multiple, $\hat{\lambda}_3 \approx 2\hat{\lambda}_1$. In such a setting, the grid fields of the larger module are a subset of the grid fields of the smaller module, up to a rotation (due to the difference in orientation between the two modules), and we again find that variability in grid properties can improve decoding accuracy (*Figure 6—figure supplement 2B*, compare black and blue lines). Similar results are found when using experimentally found values of $\hat{\lambda}$ (*Stensola et al., 2012*) and not assuming an exact $\hat{\lambda}_{n+1} = \sqrt{2}\hat{\lambda}_n$ relationship (*Figure 6—figure supplement 2C, D*).

Taken together, these results suggest that *individual* grid modules can exhibit significant encoding of local space via heterogeneity in their grid properties, even when the extent of the variability in $\theta$ and $\lambda$ is similar to that found in the analysis of the experimental recordings. This benefit can also improve encoding in cases when multiple, non-consecutive modules are considered.

## Discussion

The multiple firing fields of grid cells, organized along a triangular lattice, has been historically interpreted as a limiting feature for the encoding of local space. Particularly influential in shaping this view has been the discovery of the distribution of grid cells into distinct modules, with grid cell spacing ($\lambda$) and orientation ($\theta$) preserved within, but not across, modules (*Stensola et al., 2012*; *Gu et al., 2018*), making integration across multiple modules necessary for spatial information to be decoded (*Fiete et al., 2008*; *Sreenivasan and Fiete, 2011*; *Mathis et al., 2012a*; *Wei et al., 2015*; *Stemmler et al., 2015*). While evidence for discontinuity in the grid cell properties across modules is strong, the corollary assumption, that within-module values of $\lambda$ and $\theta$ are identical (up to the bounds of noise), has not been systematically studied.

Analyzing recently collected MEC recordings, we found the range of $\lambda$ and $\theta$ values was large, with examples of grid cell pairs in the same module having over 5° difference in grid orientation and 20 cm difference in grid spacing (*Figure 2*). Statistical analysis shows that the variability is more robust than expected from noise, for the majority of grid cells (*Figure 5*). This was despite the fact that we used a very conservative criteria for assessing whether a grid cell was consistent enough to be included in the analysis.

Our comparison of within- and between-cell grid property variability was key to our argument, as it was for previous work using it to study the robustness of differences in peak grid field firing rates (*Dunn et al., 2017*). The absence of its use in the characterization of distribution of $\lambda$ and $\theta$ (*Yoon et al., 2013*) may be why the consistency of this heterogeneity was not identified until now. We find that our conclusion holds when performing a number of control experiments (*Figure 2—figure supplement 1*, *Figure 3—figure supplement 3*, *Figure 3—figure supplement 4*, *Figure 3—figure*

*supplement 5*, *Figure 3—figure supplement 6*) and whether we treat each grid field independently (*Figure 3*) or take the average across grid fields (*Figure 4*). This challenges the assumption that the variability observed in the grid orientation and spacing is attributable solely to measurement noise.

We find that normative recurrent neural network models that develop grid cells when optimized to perform path integration (*Cueva and Wei, 2018*; *Banino et al., 2018*; *Sorscher et al., 2019*) develop similar amounts of grid property variability (*Figure 3—figure supplement 7*). This illustrates the ability of these RNN models to capture aspects of grid cell properties that have not been previously studied, and may provide another explanation for why these models have greater similarity to real MEC recordings than other models (*Nayebi et al., 2021*). Probing these computational models in greater depth may enable a more detailed understanding of the observed grid property heterogeneity.

We hypothesized that this variability may be used to increase the fidelity at which individual grid modules can encode local space. This idea is consistent with a large body of literature showing that heterogeneity in the responses of populations of neurons increases the robustness of encoding (*Shamir and Sompolinsky, 2006*; *Chelaru and Dragoi, 2008*; *Gjorgjieva et al., 2016*; *Perez Nieves et al., 2021*). We find, in noisy synthetic populations of grid cells, that a level of variability in grid properties similar to what is quantified in the real data can be sufficient to accurately decode information of local space (*Figure 6*). This benefit is increased with larger numbers of grid cells in the population (*Figure 6D*) and is observed over a range of values for the underlying variability (*Figure 6F*). We find that the improvement was most pronounced in modules with small grid spacing (*Figure 6—figure supplement 1A*), although larger modules can see a decrease in decoding error for amounts of variability consistent with was was found in the analyzed data (*Figure 6—figure supplement 1B*).

We note that our results are additionally aligned with recent findings of heterogeneity in maximal firing rates across individual grid fields (*Diehl et al., 2017*; *Ismakov et al., 2017*; *Dunn et al., 2017*) and grid sheering (*Derdikman et al., 2009*; *Krupic et al., 2015*; *Ginosar et al., 2023*). Our work further demonstrates that, even in the absence of these perturbations, individual grid modules may encode considerably more local spatial information than previously believed.

Finally, models of the formation of orientation maps in visual cortex have demonstrated that slight angular offsets of retinal mosaics, along which retinal receptive fields are organized, can generate complex patterns (*Paik and Ringach, 2011*) similar to those found in visual cortex orientation maps (*Blasdel and Salama, 1986*). Our results indicate that grid cells in MEC may take advantage of a similar computational principle, suggesting that mosaic patterns might be a broadly utilized feature of neural coding (*Smith and Smith, 2011*).

## Limitations

While the data set we analyzed (*Gardner et al., 2022*) represents an advance in the ability to simultaneously record from tens to hundreds of putative grid cells, across grid modules, the MEC remains a challenging brain region to access for large-scale neurophysiological experiments. Indeed, with our conservative inclusion criteria, we were ultimately limited by having only 420 grid cells included in our analysis. Future work can perform more detailed and complete characterizations of grid property heterogeneity, as new neurotechnologies that enable larger yield of grid cells are developed (*Low et al., 2014*; *Zong et al., 2022*).

Our decoding analysis, while demonstrating the possibility that variability in grid properties can be used by individual grid modules to enhance the encoding of local spatial information, made several simplifying assumptions, including: (1) independent Poisson noise for neural activity, (2) linear decoding, and (3) normal distribution of grid properties. Although comparison to real data showed that these assumptions are reasonable (*Figure 6E*), future work can assess the extent to which these restrictions can be lifted (e.g. to incorporate correlated neural noise), while still enabling individual grid modules to have low decoding error.

## Open questions

The heterogeneity in grid properties we characterize motivates the investigation of several new lines of research. Because these are directions that we believe to be fruitful for the field as a whole, we outline them below, with our hypotheses for possible answers.

## Q1: How does grid property variability affect continuous attractor network structure?

Continuous attractor network models of grid cells (*Fuhs and Touretzky, 2006*; *Guanella and Verschure, 2006*; *Burak and Fiete, 2009*; *Couey et al., 2013*) enjoy considerable experimental support (*Yoon et al., 2013*; *Dunn et al., 2015*; *Gu et al., 2018*; *Gardner et al., 2022*; *Gardner et al., 2019*; *Trettel et al., 2019*), making them one of the 'canonical' models in neuroscience. However, these models make use of the assumption that all the grid cells in a given module have the same grid orientation and spacing to simplify the network connectivity. In particular, by assuming equal grid orientation and spacing, it becomes possible to arrange grid cells in a two-dimensional space spanned by their phases (i.e. a 'neural sheet' *Burak and Fiete, 2009*). Neurons close in this space are wired with excitatory connections and neurons far in this space are wired with inhibitory connections. As the data set we analyzed was found by others to provide strong support for the basic predictions of continuous attractor networks (i.e. toroidal topology of the activity manifold; *Gardner et al., 2022*), we do not view our results as directly challenging these models. That the RNN models we investigate (*Figure 3—figure supplement 7*) are explicitly trained to perform path integration (and achieve highly accurate performance *Sorscher et al., 2023*) supports our hypothesis that variability in grid properties does not necessarily destroy the continuous attractor or path integration capabilities of the MEC. Understanding how this is possible is an exciting future direction, and use of geometric (*Acosta et al., 2023*) and dynamical systems based tools (*Redman et al., 2022a*; *Redman et al., 2023*; *Ostrow et al., 2024*) may shed new light on this. We hypothesize that the degree in variability of grid spacing and orientation may strike a balance between being small enough to keep the continuous attractor network structure stable, but large enough to enable encoding of local information of space.

## Q2: What causes grid property variability?

A natural direction to address is identifying the source of the heterogeneity in grid cell properties we observe. One hypothesis is that this could be driven by 'defects' in the specific connectivity pattern that is needed for a continuous attractor. This could be due to synaptic connections between grid cells and other functional classes of neurons in the MEC, such as border (*Solstad et al., 2008*), band (*Krupic et al., 2012*), and non-spatial (*Diehl et al., 2017*) cells. Because the path integrating RNN models have been found to develop analogous responses to these classes (*Banino et al., 2018*; *Cueva and Wei, 2018*; *Schøyen et al., 2023*; *Pettersen et al., 2024a*; *Pettersen et al., 2024b*), as well as capture general properties of MEC activity (*Nayebi et al., 2021*), we examined the affect of 'connectivity noise' in RNNs. In particular, we swept across different strengths of weight decay regularization, which controls the amount of sparsity in the recurrent layer. We find that smaller weight decay (and thus, denser connectivity between units) led to greater variability in grid properties (*Figure 3—figure supplement 7E, F*). Increasing the weight decay (and thus, enforcing sparser connectivity) led to a reduction of grid property variability (*Figure 3—figure supplement 7E, F*). This supports the hypothesis that the heterogeneity in grid properties we observe may be due to 'non-perfect' connectivity.

Alternatively, the coupling between hippocampus and MEC, which has been shown to lead to variability in grid field firing rates (as experimentally observed; *Dunn et al., 2017*; *Agmon and Burak, 2020*), may lead to differences in grid orientation and spacing. In particular, a previous computational model that learned grid cells from place cell input, using non-negative principal component analysis, has shown that place field width affects grid cell orientation and spacing (*Dordek et al., 2016*). Heterogeneity in the spatial coding properties of place cells has been found along the transverse axis of CA3 (*Lee et al., 2015*; *Lu et al., 2015*; *Redman et al., 2022b*), suggesting there may be a systematic differences in the place field widths of hippocampal inputs to MEC grid cells. Further, it was shown that this place-to-grid cell computational model has a linear relationship between place field width and grid spacing, and a non-monotonic relationship between place field width and grid orientation (*Dordek et al., 2016*). These may explain why we find stronger average variability in grid spacing than grid orientation (*Figure 5A and B*).

## Q3: How does grid property variability shape hippocampal representations?

The projections from MEC to hippocampus suggest that the variability in grid properties may influence hippocampal representations (even if grid cells do not comprise the majority of its inputs *Diehl et al., 2017*). We consider two possible ways in which this may happen. First, given that grid cells

have been reported to maintain their grid spacing and orientation across exposures to new environments, while undergoing a change in their grid phase (*Fyhn et al., 2007*; *Yoon et al., 2013*), the integration across multiple modules has been necessary to explain place field remapping. However, grid phase plays an important role in generating the specific complex interference patterns that emerge when considering the joint activity of grid cells with variable grid properties (*Figure 1*). Thus the reported changes in phase may be sufficient to generate large (and seemingly random) changes in local spatial information conveyed by grid cells to hippocampal cells. This could drive additional changes in the local spatial information projected to hippocampus, as well as explain the significant differences in correlation structure between CA1 neurons across different environments (*Levy et al., 2023*).

And second, recent work on hippocampal place field drift (*Mankin et al., 2012*; *Ziv et al., 2013*; *Hainmueller and Bartos, 2018*; *Gonzalez et al., 2019*; *Dong et al., 2021*) has demonstrated that there is a significant change in the place field location across time, especially in CA1. One possible source of this phenomenon is the reported instability of dendritic spines on the apical dendrites of CA1 place cells (*Mizrahi et al., 2004*; *Attardo et al., 2015*; *Pfeiffer et al., 2018*; *Redman et al., 2022b*), ostensibly leading to changes in the MEC inputs to these neurons. However, if grid cells across multiple modules are necessary for local spatial information, turnover in synaptic input is unlikely to cause large changes in the spatial preferences of CA1 neurons, as integration over several scales should provide stable encoding properties. In contrast, different subpopulations of grid cells with variable grid properties can lead to differences in the local spatial information encoded by their joint activity, even if they come from a single module, possibly influencing the spatial preferences of CA1 place cells.

## Q4: Why are there multiple modules?

Given that our results demonstrate the ability of single grid modules to encode information about local space – a feat previously believed to be possible only if activity from multiple grid modules was integrated together – why is does MEC have multiple modules? While the variability removes the necessity for encoding local space with multiple modules, higher fidelity representations is achievable by integrating across multiple modules (*Figure 6—figure supplement 2*). In addition, the use of grid cells beyond spatial navigation (*Killian et al., 2012*; *Constantinescu et al., 2016*; *Aronov et al., 2017*; *Julian et al., 2018*; *Nau et al., 2018*; *Wilming et al., 2018*; *Neupane et al., 2024*), where hierarchical representations are important (*Klukas et al., 2020*; *Rueckemann et al., 2021*), may be a sufficient implicit bias for the formation of multiple modules (*Schaeffer et al., 2024*). In particular, encoding information at multiple distinct scales is critical for multi-scale reasoning, a cognitive function grid cells may support.

## Materials and methods
### Electrophysiology recordings

The neural activity analyzed in this paper comes from a publicly available data set, which has previously been described in detail (*Gardner et al., 2022*). We provide brief summary of the methodology and the experimental paradigms used during the recordings.

Three male rats (Long Evans – Rats Q, R, and S) were implanted with Neuropixels silicon probes (*Jun et al., 2017*; *Steinmetz et al., 2021*). These probes were targeted at the MEC-parasubiculum region and the surgery was performed as described previously (*Gardner et al., 2019*; *Steinmetz et al., 2021*). After three hours of recovery, recordings were performed.

In the recordings analyzed, the rats foraged for randomly dispersed corn puffs in a 1.5 × 1.5m$^2$ square open field arena, with walls of height 50 cm. The rats were familiar with the environment and task, having trained 10–20 times prior to the implantation. The rats were food restricted to motivate their foraging, being kept at a minimum of 90% of their original body weight (300–500 g).

All procedures in the original study were approved by the Norwegian Food and Safety Authority and done in accordance with the Norwegian Animal Welfare Act and the European Convention for the Protection of Vertebrate Animals used for Experimental and Other Scientific Purposes. Protocols were approved by the Norwegian Food Safety Authority (FOTS ID 18011 and 18013).

## Electrophysiology post-processing

The neural activity analyzed in this paper was post-processed, before made publicly available. We describe, in brief, the post-processing performed (*Gardner et al., 2022*), as well as the post-processing we performed on the downloaded data.

Spike sorting, via KiloSort 2.5 (*Steinmetz et al., 2021*), was applied to the data recorded from the Neuropixel probes. Individual units were deemed putative cells if their average spike rate was in the range of 0.5–10 Hz, and 99% of their interspike intervals were greater than 2ms.

For each putative cell, rate maps were constructed by averaging the activity at binned spatial positions in the open field arena. This raw rate map was smoothed, using a Gaussian kernel. The auto-correlation of these rate maps were computed, and a grid score calculated, as described previously (*Sargolini et al., 2006*).

From the downloaded spike trains, we constructed rate maps and autocorrelograms in a similar manner, using code made publicly available (*Banino et al., 2018*; *Figure 2*).

## Grid module classification

The public data set we analyzed (*Gardner et al., 2022*) contained the module identity of all putative grid cells. In brief, these module identities were assigned by first projecting the 2D autocorrelogram of every recorded unit onto a 2D space using the non-linear dimensionality reduction algorithm UMAP (*McInnes et al., 2018*). Then, the DBSCAN algorithm (*Ester et al., 1996*) was used to cluster the units, based on their position in the dimensionally reduced space. Cluster membership served as the basis for grid module classification, with the largest cluster being removed as it was found to not contain spatially selective autocorrelograms. This non-supervised module assignment yielded clusters of high grid scores and similar grid spacing and orientation within each cluster-module. More details can be found in *Gardner et al., 2022*. We performed no further analysis regarding module identity.

## Computing grid score, orientation, and spacing from the spatial autocorrelogram

Grid spacing and grid orientation were computed according to standard methods described in detail previously (*Stensola et al., 2012*; *Butler et al., 2019*). Briefly, the goal of the procedure is to identify the location of the six nearest fields in the spatial autocorrelogram (SAC; *Figure 2A*). This was achieved by performing the following steps. First, the SAC was smoothed using a 2D Gaussian filter with $\sigma = 1$. Second, the center of the SAC was excluded by applying a mask. Because the size of the SAC's central peak changes for every module, the radius of such mask was re-computed for each module. Third, we thresholded the SAC by applying an extended-maxima transform `ndimage.maximum_filter`. Fourth, we identified the center of every field by using the function `scipy.stats.find_peaks`.

Once the peaks of every field had been found, we computed the location of every peak in polar coordinates. We then selected the 6 peaks that were closest to the center of the SAC, based on the computed radial components. Because every SAC is symmetric, we considered for further analysis the 3 peaks closest to the X axis in angular distance (Axis 1, 2, and 3 *Stensola et al., 2015*). Grid spacing was computed as the arithmetic mean of the radial component of the 3 peaks (except when each peak was analyzed separately – *Figure 4*). Given that the SAC dimensions are twice of that of the real arena, we multiplied the SAC radial mean by a factor of 2. Grid orientation was computed as the angular mean of orientations (relative to the x-axis) of the three peaks (except when each peak was analyzed separately – *Figure 4*).

In order to ensure that subsequent analysis was performed only on cells whose SAC's could be well described by hexagonal structure, we imposed the following constraints: (1) the relative angle between two peaks could not be < 30°; (2) the relative angle between two peaks could not be > 90°; (3) the values for grid spacing between the peaks could not be substantially different (the ratio of spacings between any two peaks must be > 0.5 and < 2). Cells that did not meet these criteria were determined to have "Poor grid fit" and were rejected from all subsequent analysis. The percentage of all cells that were removed by this inclusion criteria is shown in *Figure 3—figure supplement 1*. We note that cells from one of the nine modules in the publicly available data set that we analyzed had 98.4% of cells rejected by this criteria (recording identifier – R23). We therefore did not include it in any subsequent analysis.

To compute the grid score of recorded MEC cells, we made use of previously published code (*Banino et al., 2018*), that is based on metrics that have become standards in quantifying grid cell properties (*Sargolini et al., 2006*).

For analysis of the distribution of grid properties (*Figure 2E and F*), we included only grid cells with grid scores greater than 0.85. This was done to demonstrate that variability was present even in cells that exhibit robust grid cell properties. In the subsequent analyses, an alternative criteria is used, which considers the reliability of the grid responses (see below).

## Computing spacing from the rate maps

In order to characterize the spacing with a method that is less susceptible to the effects of having grid fields at the boundaries of the environment (as the SAC method could be), we compute the spacing using the rate maps of individual grid cells (*Figure 2—figure supplement 1A*). Once the rate map was computed, we smoothed it with a Gaussian filter, setting $\sigma = 1.5$. Then, using the function `scikit-image.feature.peak_local_max` we extract the position of the center of each firing field. Because our goal with this analysis is to show that the fields in the border do not impact our analysis with the SAC, we restrict ourselves to the 3 peaks that are closest to the center of the arena (*Figure 2—figure supplement 1A*). We compute the distances between those three peaks to each other and report the spacing as the average of the distances measured.

## Within and between cells splits

To characterize the within- and between-cell variability of grid spacing and orientation, we employed the following approach. First, we split the data into bins of fixed length (30 s). From this, we randomly assigned each interval to one of two blocks (denoted as blocks *A* and *B*), with exactly half the total number of intervals in each block. For each grid cell, we computed the grid spacing [$\lambda_A^{(i)}$ and $\lambda_B^{(i)}$] and orientation [$\theta_A^{(i)}$ and $\theta_B^{(i)}$], from the data in each block. The within-cell differences of grid spacing and orientation was determined as $\Delta\theta_{\text{within}}^{(i)} = \theta_A^{(i)} - \theta_B^{(i)}$ and $\Delta\lambda_{\text{within}}^{(i)} = \lambda_A^{(i)} - \lambda_B^{(i)}$. Each grid cell's properties were also compared those of another grid cell, with the match being made using random sampling without replacement. These comparisons were determined as $\Delta\theta_{\text{between}}^{(i,j)} = \theta_A^{(i)} - \theta_B^{(j)}$ and $\Delta\lambda_{\text{between}}^{(i,j)} = \lambda_A^{(i)} - \lambda_B^{(j)}$. This process was repeated 100 times (referring to each iteration as a 'shuffle'), per recording. Examples of splits of the data, for different shuffles, is schematically illustrated in *Figure 3A*. The resulting distributions of $\Delta\theta_{\text{within}}$, $\Delta\lambda_{\text{within}}$, $\Delta\lambda_{\text{within}}$, and $\Delta\lambda_{\text{between}}$, across grid cells and splits of the data were then compared. If the within-cell variability in grid spacing and orientation was smaller than between-cell variability, we concluded that the variability of grid cell properties was a robust feature of the data and not due to noise.

When performing this shuffle analysis, we found that some cells, despite having a good grid fit when all the data was considered, did not have SACs that were well described by hexagonal structure when the data was split in half. We viewed this a manifestation of unreliable grid coding and a possible confound in our quantification of variability. As such, we introduced a new inclusion criteria (replacing that of requiring a grid score of > 0.85), only considering cells that had poor grid fits on < 5% of all shuffles. The percentage of all cells that were removed by this inclusion criteria is shown in *Figure 3—figure supplement 1*. In general, we found that cells with high grid score were reliable, although there were exceptions. Additionally, we found that size of the bin used for splitting the data did not significantly affect the percent of cells with good grid fits (passed the prior inclusion criteria) that were considered reliable (*Figure 3—figure supplement 2*).

## Synthetic grid cells

To study how variability in grid cell properties might endow the grid code with computational advantages, we generated synthetic grid cell rate maps, so that we could have complete control over the distribution of their properties. These synthetic grid cell rate maps were constructed as follows.

First, the lengths of each dimension the 'arena' within which the simulated grid cells exist ($L_x$ and $L_y$) were set. Then, for $N \in \mathbb{N}$ grid cells, the grid spacing and orientation were sampled via $\lambda^{(i)} \sim \mathcal{N}(\mu_\lambda, \sigma_\lambda)$ and $\theta^{(i)} \sim \mathcal{N}(\mu_\theta, \sigma_\theta)$ where $\mathcal{N}(\mu, \sigma)$ is a normal distribution with mean μ and variance $\sigma^2$. Grid phase was sampled as $\phi^{(i)} \sim \mathcal{U}([0, L_x] \times [0, L_y])$, where $\phi$ is a two dimensional vector, with first component uniformly sampled from $[0, L_x]$ and second component uniformly sampled from $[0, L_y]$. To

construct a population with no variability in grid properties (to use as a control), we set $\sigma_\lambda = 0$ m. and $\sigma_\theta = 0°$.

For each grid cell, we generated idealized grid responses by summing three two-dimensional sinusoids (**Solstad et al., 2006**), such that the activity at $\mathbf{x} = (x, y) \in [-L_x/2, L_x/2] \times [-L_y/2, L_y/2]$ is given by

$$X^{(i)}(\mathbf{x}) = X_i^{\max} \frac{2}{3} \left( \frac{1}{3} \sum_{j=1}^{3} \cos \left[ \mathbf{k}_j(\theta_i)(\mathbf{x} + \phi_i) \right] + \frac{1}{2} \right),$$ (1)

where $X_i^{\max}$ is the maximal firing rate and $\mathbf{k}_j(\theta_i)$ are the wave vectors with 0°, 60°, and 120° angular differences

$$
\begin{aligned}
k_1(\theta_i) &= \frac{k}{\sqrt{2}} \cdot [\cos(\theta_i + \pi/12) + \sin(\theta_i + \pi/12), \cos(\theta_i + \pi/12) - \sin(\theta_i + \pi/12)] \\
k_2(\theta_i) &= \frac{k}{\sqrt{2}} \cdot [\cos(\theta_i + 5\pi/12) + \sin(\theta_i + 5\pi/12), \cos(\theta_i + 5\pi/12) - \sin(\theta_i + 5\pi/12)] \\
k_3(\theta_i) &= \frac{k}{\sqrt{2}} \cdot [\cos(\theta_i + 3\pi/4) + \sin(\theta_i + 3\pi/4), \cos(\theta_i + 3\pi/4) - \sin(\theta_i + 3\pi/4)],
\end{aligned}
$$ (2)

where $k = 4\pi/(\sqrt{3}\lambda_i)$.

To match the recorded neural data, where individual grid cells have distinct maximal firing rates, $X_i^{\max} \sim \mathcal{N}(13, 8)$. We enforced $X_i^{\max}$ to be within $[2, 30]$, by setting any sampled values outside of this range to the boundary values (i.e. 2 or 30).

To determine the extent to which local spatial information can be decoded from the activity of populations of grid cells with different degrees of variability in their grid properties, we performed the following analysis.

We generated noisy synthetic spike rates of $N$ grid cells by assuming a Poisson process and sampling using the idealized rate maps (**Equation 1**). More concretely, the activity of grid cell $i$ at position $(x, y)$ was assumed to be a random variable with a Poisson distribution, whose mean was $X^{(i)}(x, y)$. Thus, the probability of observing $\tilde{X}^{(i)}(x, y)$ spikes, at position $(x, y)$, is given by

$$\tilde{X}^{(i)}(x, y) \sim \mathcal{P}\left( X^{(i)}(x, y) \right).$$ (3)

## Linear decoding synthetic data

For a given resolution of the arena, we generated $\tilde{X}_j^{(i)}(x, y)$, where $j = 1, ..., 10$. That is, we constructed 10 noisy rate maps. We performed cross-validated decoding by averaging across 9 of the 10 rate maps, to get an average rate map $\hat{X}^{(i)}(x, y)$. For sake of simplicity, consider $\hat{X}^{(i)}(x, y) = 1/9 \sum_{j=1}^{9} \tilde{X}_j^{(i)}(x, y)$.

To decode local position from the held out noisy rate map [e.g. $\tilde{X}_{10}^{(i)}(x, y)$], we multiplied the activity at each position by all the positions in the average rate map, taking the sum and assigning the decoded position as that with the largest value. The Euclidean distance between the decoded position and the true position is considered the error. This was performed 10 times (holding out each rate map once) and the average error across all positions in the environment was then averaged across all 10 of the validation splits.

## Linear decoding experimental data

To decode the electrophysiological data (**Gardner et al., 2022**), we sampled subpopulations of grid cells from the $N = 82$ units (recording ID R12) that passed the criteria described previously. For each subpopulation, we split the recorded activity into 10 evenly spaced temporal bins, and constructed average ratemaps for each bin. This is consistent with what was done in decoding the synthetic data. All ratemaps, for each cell, were normalized by the maximal activity. We then randomly chose 5 of the 10 time bins to serve as the 'train' data, the remaining 5 served as the 'test' data. The rate maps were averaged and then the linear decoder was applied. We sampled 10 different choices of splitting the data and 25 choices of subpopulation.

**Table 1.** Parameters used to train RNNs on path integration.

See *Sorscher et al., 2019*; *Sorscher et al., 2023* for more information on these parameters.

| Parameter | Value |
| --- | --- |
| Epochs | 100 |
| Batch size | 200 |
| Batches per epoch | 1000 |
| Path length ($T$) | 20 |
| Arena length (L) | 2.2m |
| Learning rate | $10^{-4}$ |
| Place cells ($n_p$) | 512 |
| Grid cells ($n_G$) | 2096 |
| $\sigma_1$ | 0.12 |
| $\sigma_2$ | 0.24 |
| Activation | ReLU |
| Weight decay | $10^{-4}$ |
| Optimizer | Adam |

## Recurrent neural network model

Code used for training and evaluating path integrating recurrent neural network (RNNs) (*Sorscher et al., 2019*; *Sorscher et al., 2023*) was pulled from https://github.com/ganguli-lab/grid-pattern-formation (*Schaeffer et al., 2023*). We used the same hyperparameters as are present on the repository (see *Table 1*), except when examining the effect of weight regularization on grid property variability, in which case we trained RNNs with weight decay magnitude from {$10^{-6}$, $10^{-5}$, $10^{-4}$, $2.5 \cdot 10^{-4}$}. The greater the value, the greater the enforced sparsity. These were chosen as they had previously been found to lead to good path integration performance and have high grid scores (*Sorscher et al., 2023*). We independently verified that this was true for the resulting RNNs. We computed the grid score, spacing, and orientation using similar code implementations as was used for the neural data. For each value of weight decay, we trained three independent RNNs, using different random seeds.

## Acknowledgements

We thank the members of the Goard Lab, Francisco Acosta, Spencer Smith, Caleb Kemere, Will Dorrell, and the DYNS graduate students for useful discussions surrounding this work. We thank Andreas Herz for suggesting the analysis present in *Figure 2—figure supplement 1* and William Dorrell for suggesting the analysis present in *Figure 6—figure supplement 2C, D*. We thank the eLife reviewers for their suggestions. This work was supported by grants to MJG from NIH (R01 NS121919), NSF (1934288), and the Whitehall Foundation, and a grant to XXW from NSF (2318065).

## Additional information

### Funding

| Funder | Grant reference number | Author |
| --- | --- | --- |
| National Institutes of Health | R01 NS121919 | Michael J Goard |
| Whitehall Foundation | 2022-05-009 | Michael J Goard |
| National Science Foundation | 2318065 | Xue-Xin Wei |

| Funder | Grant reference number | Author |
|---|---|---|
| National Science Foundation | 1934288 | Michael J Goard |

The funders had no role in study design, data collection and interpretation, or the decision to submit the work for publication.

## Author contributions

William T Redman, Conceptualization, Software, Formal analysis, Validation, Investigation, Visualization, Methodology, Writing – original draft, Writing – review and editing; Santiago Acosta-Mendoza, Software, Formal analysis, Validation, Investigation, Visualization, Methodology, Writing – review and editing; Xue-Xin Wei, Supervision, Validation, Writing – review and editing; Michael J Goard, Conceptualization, Supervision, Funding acquisition, Writing – review and editing

## Author ORCIDs

William T Redman ⓘD https://orcid.org/0000-0002-4147-2026
Santiago Acosta-Mendoza ⓘD https://orcid.org/0009-0003-6698-476X
Michael J Goard ⓘD https://orcid.org/0000-0002-5366-8501

Reviewer #1 (Public review): https://doi.org/10.7554/eLife.100652.3.sa1
Reviewer #2 (Public review): https://doi.org/10.7554/eLife.100652.3.sa2
Reviewer #3 (Public review): https://doi.org/10.7554/eLife.100652.3.sa3
Author response https://doi.org/10.7554/eLife.100652.3.sa4

# Additional files

## Supplementary files

MDAR checklist

## Data availability

Code used for analysis is publicly available at GitHub, copy archived at *Redman, 2025*. Data was made available by the Moser Lab at figshare (*Gardner et al., 2022*).

The following previously published dataset was used:

| Author(s) | Year | Dataset title | Dataset URL | Database and Identifier |
|---|---|---|---|---|
| Gardner R, Hermansen E | 2022 | Toroidal topology of population activity in grid cells | https://doi.org/10.6084/m9.figshare.16764508 | figshare, 10.6084/m9.figshare.16764508 |

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
