## [Editor Report · eLife Assessment]

This **valuable** study examines the variability in spacing and direction of entorhinal grid cells, providing **convincing** evidence that such variability helps disambiguate locations within an environment. This study will be of interest to neuroscientists working on spatial navigation and, more broadly, on neural coding.

---

## [Referee Report · Reviewer #1 (Public review)]

Summary:

The present paper by Redman et al. investigated the variability of grid cell properties in the MEC by analyzing publicly available large-scale neural recording data. Although previous studies have proposed that grid spacing and orientation are homogeneous within the same grid module, the authors found a small but robust variability in grid spacing and orientation across grid cells in the same module. The authors also showed, through model simulations, that such variability is useful for decoding spatial position.

Strengths:

The results of this study provide novel and intriguing insights into how grid cells compose the cognitive map in the axis of the entorhinal cortex and hippocampus. This study analyzes large data sets in an appropriate manner and the results are convincing.

Comments on revisions:

In the revised version of the manuscript, the authors have addressed all the concerns I raised.

---

## [Referee Report · Reviewer #2 (Public review)]

Summary:

This paper presents an interesting and useful analysis of grid cell heterogeneity, showing that the experimentally observed heterogeneity of spacing and orientation within a grid cell module can allow more accurate decoding of location from a single module.

Strengths:

(1) I found the statistical analysis of the grid cell variability to be very systematic and convincing. I also found the evidence for enhanced decoding of location based on between cell variability within a module to be convincing and important, supporting their conclusions.

(2) Theoreticians have developed models that focus on the use of grid cells that are highly regular in their parameters, and usually vary only in the spatial phase of cells within modules and the spacing and orientation between modules. This focus on consistency is partly to obtain the generalization of the grid cell code to a broad range of previously unvisited locations. In contrast, most experimentalists working with grid cells know that many if not most grid cells show high variability of firing fields, as demonstrated in the figures in experimental papers. The authors of this current paper have highlighted this discrepancy, and shown that the variability shown in the data could actually enhance decoding of location.

---

## [Referee Report · Reviewer #3 (Public review)]

Summary:

Redman and colleagues analyze grid cell data obtained from public databases. They show that there is significant variability in spacing and orientation within a module. They show that the difference in spacing and orientation for a pair of cells is larger than the one obtained for two independent maps of the same cell. They speculate that this variability could be useful to disambiguate the rat position if only information from a single module is used by a decoder.

Strengths:

The strengths of this work lie in its conciseness, clarity, and the potential significance of its findings for the grid cell community, which has largely overlooked this issue for the past two decades. Their hypothesis is well stated and the analyses are solid.

Weaknesses:

Major weaknesses identified in the original version have been addressed.

The authors have addressed all of our concerns, providing control analyses that strengthen their claim.

---

## [Author Response]

The following is the authors’ response to the original reviews.

We thank the reviewers for their time and thoughtful comments. We believe that the further analyses suggested have made the results clearer and more robust. Below, we briefly highlight the key points addressed in the revision and the new evidence supporting them. Then, we address each reviewer’s critiques point-by-point.

- Changes in variability with respect to time/experience

Both reviewers #1 and #3 asked whether the variability in grid properties observed was dependent on time or experience. This is an important point, given that such a dependence on time could lead to interesting hypotheses about the underlying dynamics of the grid code. However, in the new analyses we performed, we do not observe changes in grid variability within a session (Fig S5 of the revised manuscript), suggesting that the grid variability seen is constant within the timescale of the data set.

- The assumption of constant grid parameters in the literature

Reviewer #2 pointed out that it had been appreciated by experimentalists that grid properties are variable within a module. We agree that we may have overstated the universality of this assumption in the original manuscript, and we have toned down the language in the revision. However, we note that many previous theoretical studies assumed these properties to be constant, within a given module. We provide some examples below, and have added evidence of this assertion, with citations to the theoretical literature, to the revised manuscript .

- Additional sources of variability

Reviewer #3 pointed out additional sources that might explain the variability observed in the paper (beyond time and experience). These sources include: field width, border location, and the impact of conjunctive cells. We have run additional analyses and have found no significant impact on the observed variability from any of these factors. We believe that these are important controls, and have added them to the manuscript (Fig S4-S7 of the revised manuscript)

- Analysis of computational models

Reviewer #3 noted that our results could be strengthened by performing similar analyses on the output of computational models of grid cells. This is a good idea. We have now measured the variability of grid properties in a recent normative recurrent neural network (RNN) model that develops grid cells when trained to perform path integration (Sorscher et al., 2019). This model has been shown to develop signatures of a 2D toroidal attractor (Sorscher et al., 2023) and achieves a high accuracy on a simple path integration task. Interestingly, the units with the greatest grid scores also exhibit a range of grid spacings and grid orientations (Fig S8 of the revised manuscript). Furthermore, by decreasing the amount of sparsity (through decreasing the weight decay regularization), we found an increase in the variability of the grid properties. This analysis demonstrates a heretofore unknown similarity between the RNN models trained to perform path integration and recorded grid cells from MEC. It additionally provides a framework for computational analysis of the emergence of grid property variability.

**Reviewer #1:**
(1) Is the variability in grid spacing and orientation that the authors found intrinsically organized or is it shaped by experience? Previous research has shown that grid representations can be modified through experience (e.g., Boccara et al., Science 2019). To understand the dynamics of the network, it would be important to investigate whether robust variability exists from the beginning of the task period (recording period) or whether variability emerges in an experience-dependent manner within a session.

This is an interesting question that was not addressed in the paper. To test this, we performed additional analysis to resolve whether the variability changes across a session.

Using a sliding window, we have measured changes in variability with respect to recording time (Fig S5A). To this end, we compute grid orientation and spacing over a time-window whose length is half the total length of the recording. From the population distribution of orientation and spacing values, we compute the standard deviation as a measure of variability. We repeat the same procedure, sliding the window forward until the variability for the second half of the recording is computed.

We applied this approach to recording ID R12 (the same as in Figs 2-4) given that this recording session was significantly longer than the rest (nearly two hours). Results are shown in Fig S5B-C. For both orientation and spacing, no changes of variability with respect to time can be observed. Similar results were found for other modules (see caption of Fig S5 for statistics).

We also note that the rats were already familiarized with the environment for 10-20 sessions prior to the recordings, so there may not be further learning during the period of the grid cell recordings. No changes in variability can be seen in Rat R across days (e.g., in Fig 5B R12 and R22 have similar distributions of variability). However, we note that it may be possible that there are changes in grid properties at time-scales greater than the recordings.

(2) It is important to consider the optimal variability size. The larger the variability, the better it is for decoding. On the other hand, as the authors state in the

Discussion, it is assumed that variability does not exist in the continuous attractor model. Although this study describes that it does not address how such variability fits the attractor theory, it would be better if more detailed ideas and suggestions were provided as to what direction the study could take to clarify the optimal size of variability.

We appreciate this suggestion and agree that more discussion is warranted on how our results can be reconciled with previously observed attractor dynamics. To explore this, we studied the recurrent neural network (RNN) model from Sorscher et al. (2019), which develops grid responses when trained on path integration. This network has previously been found to develop signatures of toroidal topology (Sorscher et al., 2023), yet we find its grid responses also contain heterogeneity in grid properties (Fig S8). By decreasing the strength of the weight decay regularization (which leads to denser connectivity in the recurrent layer), we find an increase in the grid property variability. Interestingly, decreasing the weight decay regularization has been previously found to lead to weaker grid responses and worse ability of the RNN to perform path integration on environments larger than it was trained on. This approach not only provides preliminary evidence to our claim that too much variability can lead to weaker continuous attractor structure, but also provides a modeling framework with which future work can explore this question in more detail. We have added discussion of this issue to the manuscript text (Discussion).

**Reviewer #2:**
(1) Even though theoreticians might have gotten the mistaken impression that grid cells are highly regular, this might be due to an overemphasis on regularity in a subset of papers. Most experimentalists working with grid cells know that many if not most grid cells show high variability of firing fields within a single neuron, though this analysis focuses on between neurons. In response to this comment, the reviewers should tone down and modify their statements about what are the current assumptions of the field (and if possible provide a short supplemental section with direct quotes from various papers that have made these assumptions).

We agree that some experimentalists are aware of variability in the recorded grid response patterns and that this work may not come as a complete surprise to them. We have toned down our language in the Introduction, changing “our results challenge a long-held assumption” to “our results challenge a frequently made assumption in the theoretical literature”. Additionally, we have added a caveat that “experimentalists have been aware” of the observed variability in grid properties.

We would like to emphasize that the lack of work carefully examining the robustness of this variability has prevented a firm understanding of whether this is an inherent property of grid cells or due to measurement noise. The impact of this can be seen in theoretical neuroscience work where a considerable number of articles (including recent publications) start with the assumption that all grid cells within a module have identical properties, with the exception of phase shift and noise. We have now cited a number of these papers in the Introduction, to provide specific references. To further illustrate the pervasiveness of this assumption being explicitly made in theoretical neuroscience, below we provide quotes from a few important papers:

“Cells with a common spatial period also share a common grid orientation; their responses differ only by spatial translations, or different preferred firing phases, with respect to their common response period” (Sreenivasan and Fiete, 2011)”

“Grid cells are organized into discrete modules; within each module, the spatial scale and orientation of the grid lattice are the same, but the lattice for different cells is shifted in space.” (Stemmler et al., 2015)”

“Recently, it was shown that grid cells are organized in discrete modules within which cells share the same orientation and periodicity but vary randomly in phase” (Wei et al., 2015)”

“...cells within one module have receptive fields that are translated versions of one another, and different modules have firing lattices of different scales and orientations” (Dorrell et al., 2023)”

In these works, this assumption is used to derive properties relating to the computational properties of grid cells (e.g., error correction, optimal scaling between grid spacings in different modules).

In addition, since grid cells are assumed to be identical in the computational neuroscience community, there has been little work on quantifying how much variability a given model produces. This makes it challenging to understand how consistent different models are with our observations. This is illustrated in our analysis of a recent recurrent neural network (RNN) model of grid cells (Fig S8), which does exhibit variability.

(2) The authors state that "no characterization of the degree and robustness of variability in grid properties within individual modules has been performed." It is always dangerous to speak in absolute terms about what has been done in scientific studies. It is true that few studies have had the number of grid cells necessary to make comparisons within and between modules, but many studies have clearly shown the distribution of spacing in neuronal data (e.g. Hafting et al., 2005; Barry et al., 2007; Stensola et al., 2012; Hardcastle et al., 2015) so the variability has been visible in the data presentations. Also, most researchers in the field are well aware that highly consistent grid cells are much rarer than messy grid cells that have unevenly spaced firing fields. This doesn't hurt the importance of the paper, but they need to tone down their statements about the lack of previous awareness of variability (specific locations are noted in the specific comments).

We have toned down our language in the Introduction. However, we note that our point that no detailed analysis had been done on measuring the robustness of this variability stands. Thus, for the general community, it has not been clear whether this previously observed variability is noise or a real feature of the grid code.

(3) The methods section needs to have a separate subheading entitled: How grid cells were assigned to modules" that clearly describes how the grid cells were assigned to a module i.e. was this done by Gardner et al., or done as part of this paper's post-processing?

We thank the reviewer for pointing out this missing information. We have added a new subsection in the Materials and Methods section, entitled “Grid module classification” to clarify how the grid cells are assigned to modules. In short, this was done by Gardner et al. (2022) using an unsupervised clustering approach that was viewed as enabling a less biased identification of modules. We did not perform any additional processing steps on module identity.

**Reviewer #3:**
(1) One possible explanation of the dispersion in lambda (not in theta) could be variability in the typical width of the field. For a fixed spacing, wider fields might push the six fields around the center of the autocorrelogram toward the outside, depending on the details of how exactly the position of these fields is calculated. We recommend authors show that lambda does not correlate with field width, or at least that the variability explained by field width is smaller than the overall lambda variability.

We agree that this option had not been carefully ruled out by our previous analyses. To tackle this question, we compute the field width of a given cell using the value at the minima of its spatial autocorrelogram (Fig S4A-B). For all cells in recording ID R12, there is a non-significant negative linear correlation between grid field width and between-cell variability (Fig S4C) . The variability explained by the width of the field is 4% of the variability, as indicated by the R^2^ value of the linear fit. Similar results were found for all other modules (see caption of Fig S4C for statistics). Therefore, we do not think that grid field width explains spacing variability.

(2) An alternative explanation could be related to what happens at the borders. The authors tackle this issue in Figure S2 but introduce a different way of measuring lambda based on three fields, which in our view is not optimal. We recommend showing that the dispersions in lambda and theta remain invariant as one removes the border-most part of the maps but estimating lambda through the autocorrelogram of the remaining part of the map. Of course, there is a limit to how much can be removed before measures of lambda and theta become very noisy.

We have performed additional analysis to explore the role of borders in grid property variability. To do so, we have followed the suggestion by the reviewer and have re-analyzed grid properties from the autocorrelogram when the border-most part of the maps are removed (Fig S6A-B). For all modules, we do not see any changes in variability (computed as the standard deviation of the population distribution) for either orientation or spacing. As predicted by the reviewer, after removing about 25% of the border-most part of the environment we start seeing changes in variability, as measures of theta and lambda become noisy and computed over a smaller spatial range. This result holds for all other modules (Fig S6C-D).

(3) A third possibility is slightly more tricky. Some works (for example Kropff et al, 2015) have shown that fields anticipate the rat position, so every time the rat traverses them they appear slightly displaced opposite to the direction of movement. The amount of displacement depends on the velocity. Maps that we construct out of a whole session should be deformed in a perfectly symmetric way if rats traverse fields in all directions and speeds. However, if the cell is conjunctive, we would expect a deformation mainly along the cell's preferred head direction. Since conjunctive cells have all possible preferred directions, and many grid cells are not conjunctive at all, this phenomenon could create variability in theta and lambda that is not a legitimate one but rather associated with the way we pool data to construct maps. To rule away this possibility, we recommend the authors study the variability in theta and lambda of conjunctive vs non-conjunctive grid cells. If the authors suspect that this phenomenon could explain part of their results, they should also take into account the findings of Gerlei and colleagues (2020) from the Nolan lab, that add complexity to this issue.

We appreciate the reviewer pointing out the possible role conjunctive cells may play. To investigate how conjunctive cells may affect the observed grid property variability, we have performed additional analyses taking into account if the grid cells included in the study are conjunctive. Comparing within- and between-cell variability of conjunctive vs. non-conjunctive cells in recording R12, we do not see any qualitative differences for either orientation or spacing (Fig S7A-B). When excluding conjunctive cells from the between-variability comparison, we do not see any significant difference compared to when these cells are included (Fig S7C-D). As such, it does not appear that conjunctive cells are the source of variability in the population.

We further note that the number of putative conjunctive cells varied across modules and recordings. For instance, in recording Q1 and Q2, Gardner et al. (2022) reported 3 (out of 97) and 1 (out of 66) conjunctive cells, respectively. Given that we see variability robustly across recordings (Fig 5), we do not believe that conjunctive cells can explain the presence of variability we observe.

(4) The results in Figure 6 are correct, but we are not convinced by the argument. The fact that grid cells fire in the same way in different parts of the environment and in different environments is what gives them their appeal as a platform for path integration since displacement can be calculated independently of the location of the animal. Losing this universal platform is, in our view, too much of a price to pay when the only gain is the possibility of decoding position from a single module (or non-adjacent modules) which, as the authors discuss, is probably never the case. Besides, similar disambiguation of positions within the environment would come for free by adding to the decoding algorithm spatial cells (non-hexagonal but spatially stable), which are ubiquitous across the entorhinal cortex. Thus, it seems to us that - at least along this line of argumentation - with variability the network is losing a lot but not gaining much.

We agree that losing the continuous attractor network (CAN) structure and the ability to path integrate would be a very large loss. However, we do not believe that the variability we observe necessarily destroys either the CAN or path integration. We argue this for two reasons. First, the data we analyzed [from Gardner et al. (2022)] is exactly the data set that was found to have toroidal topology and therefore viewed to be consistent with a major prediction of CANs. Thus, the amount of variability in grid properties does not rule out the underlying presence of a continuous attractor. Second, path integration may still be possible with grid cells that have variable properties. To illustrate this, we analyzed data from Sorscher et al. (2019) recurrent neural network model (RNN) that was trained explicitly on path integration, and found that the grid representations that emerged had variability in spacing and orientation (see point #6 below).

(5) In Figure 4 one axis has markedly lower variability. Is this always the same axis? Can the authors comment more on this finding?

We agree that in Fig 4 the first axis has lower variability. We believe that this is specific to the module R12 and does not reflect any differences in axis or bias in the methods used to compute the axis metrics. To test this, we have performed the same analyses for other modules, finding that other recordings do not exhibit the same bias. Results for the modules with the most cells are shown below (Author response image 1).

**Author response image 1. sa4fig1:** Grid propertied along Axis 1 are not less variable for many recorded grid modules. Same as Fig.4C-D, but for four other recorded modules. Note that the variability along each axis is similar.

(6) The paper would gain in depth if maps coming out of different computational models could be analyzed in the same way.

We agree with the reviewer that examining computational models using the same approach would strengthen our results and we appreciate the suggestion. To address this, we have analyzed the results from a previous normative model for grid cells [Sorscher et al., (2019)] that trained a recurrent neural network (RNN) model to perform path integration and found that units developed grid cell like responses. These models have been found to exhibit signatures of toroidal attractor dynamics [Sorscher et al. (2023)] and exhibit a diversity of responses beyond pure grid cells, making them a good starting point for understanding whether models of MEC may contain uncharacterized variability in grid properties.

We find that RNN units in these normative models exhibit similar amounts of variability in grid spacing and orientation as observed in the real grid cell recordings (Fig S8A-D). This provides additional evidence that this variability may be expected from a normative framework, and that the variability does not destroy the ability to path integrate (which the RNN is explicitly trained to perform).

The RNN model offers possibilities to assess what might cause this variability. While we leave a detailed investigation of this to future work, we varied the weight decay regularization hyper-parameter. This value controls how sparse the weights in the hidden recurrent layer are. Large weight decay regularization strength encourages sparser connectivity, while small weight decay regularization strength allows for denser connectivity. We find that increasing this penalty (and enforcing sparser connectivity) decreases the variability of grid properties (Fig S8E-F). This suggests that the observed variability in the Gardner et al. (2022) data set could be due to the fact that grid cells are synaptically connected to other, non-grid cells in MEC.

(7) Similarly, it would be very interesting to expand the study with some other data to understand if between-cell delta_theta and delta_lambda are invariant across environments. In a related matter, is there a correlation between delta_theta (delta_lambda) for the first vs for the second half of the session? We expect there should be a significant correlation, it would be nice to show it.

We agree this would be interesting to examine. For this analysis, it is essential to have a large number of grid cells, and we are not aware of other published data sets with comparable cell numbers using different environments.

Using a sliding window analysis, we have characterized changes in variability with respect to the recording time (Figure S5A). To do so, we compute grid orientation and spacing over a time-window whose length is half of the total length of the recording. From the population distribution of orientation and spacing values, we compute the standard deviation as a measure of between-cell variability. We repeat the same procedure, sliding the window forward until the variability for the second half of the recording is computed.

We applied this approach to recording ID R12 (the same as in Figs 2-4) given that this recording session was significantly longer than the rest (almost two hours). Results are shown in Fig S5 B-C. For both orientation and spacing, no systematic changes of variability with respect to time were observed. Similar results were found for other modules (see caption of Fig S5 for statistics).

We also note that the rats were already familiarized with the environment for 10-20 sessions prior to the recordings, so there may not be further learning during the period of the grid cell recordings. No changes in variability can be seen in Rat R across days (e.g., in Fig 5B R12 and R22 have similar distributions of variability). However, we note that it may be possible that there are changes in grid properties at time-scales greater than the recordings.